



# Aeroelastic response of a multi-megawatt upwind HAWT based on fluid-structure interaction simulation

Yasir Shkara, Martin Cardaun, Ralf Schelenz and Georg Jacobs

Center for Wind Power Drives, RWTH Aachen University, Aachen, 52074, Germany

*Correspondence to*: Yasir Shkara (yasir.shkara@cwd.rwth-aachen.de)

**Abstract.** With the increase demand for greener, sustainable and economical energy sources, wind energy has proven a potential promising sustainable source of energy. The trend development of wind turbines tends to increase rotor diameter and tower height to capture more energy. The bigger, lighter and more flexible structure is more sensitive to smaller excitations. To make sure that the dynamic behavior of the wind turbine structure will not influence the stability of the system and to

further optimize the structure, a fully detailed analyses of the entire wind turbine structure is crucial.

Since the fatigue and the excitation of the structure are highly depend on the aerodynamic forces, it is important to take blade-tower interaction into consideration in the design of large-scale wind turbines. In this work, an aeroelastic model that describes the interaction between the blade and the tower of a horizontal axis wind turbine (HAWT) is presented. The high-fidelity fluid-structure interaction (FSI) model is developed by coupling a computational fluid dynamics (CFD) solver with finite element

(FE) solver to investigate the response of a multi-megawatt wind turbine structure. The results of the computational simulation showed that the dynamic response of the tower is highly depend on the rotor azimuthal position. Furthermore, rotation of the blades in front of the tower cause not only aerodynamic force pulls on the blade but a sudden reduction of the rotor aerodynamic torque by 2.3 % three times per revolution.

## 1 Introduction

Wind energy is an abundant energy source compared to other rentable energy resources. Today, multi-megawatt wind turbine is more powerful and sophisticated than the early versions. Designers have optimized wind turbines making them more efficient, cheaper and more competitive to other renewable energy generators.

It is important that the wind turbine operate in a stable condition to avoid structure vibration. In most cases, the structure absorbs the input energy leading to decrease of vibration amplitude. However, under-estimated or neglected aerodynamic-

structure interaction can lead to energized violent vibration that ended with serious structure fatigue damages. Accordingly, the importance of fatigue in the design of a wind turbine is higher than other rotary machines for a life time in the range of 20 – 30 years.



## 1.1 Horizontal axis wind turbine structure

Horizontal axis wind turbine can be described as a low stiffness dynamic system comprise complex interaction between its individual components and the surrounding atmosphere. Wind turbine support structure is a long cylinder column, where the rotor and the other components are mounted at the top. The importance of the support structure is based on the facts that the tower is the most expensive part of the machine (counts to 26% of the total cost (EWEA, 2007)). In addition, the support structure must sustain the loads that occurred during the operation and to be capable to satisfy the safety of the structure for the designed life time.

Tubular tower is designed in two ways, stiff or soft. Stiff towers have a natural frequency higher the blade passing frequency, contrarily soft tower has to endure turbine vibration that make it suffer higher stress levels. Due to the variety dynamic loads that the wind turbine subjected to (e.g. erratic wind gusts, storms, rotor dynamics) cyclic loads are induced which are three dimensional in nature. Therefore, the tower structure is sensitive to vibrate under various atmospheric conditions and its own dynamics. The design / development trend of the horizontal axis wind turbines towards Low-cost, Large-scale wind turbines. Not only increasing the rotor diameter will raise the turbine power, but duplicate wind velocity will boost the power eight times. For these reasons and in addition to wind shear, it makes sense to increase tower height so that more energy can be captured.

## 1.2 Challenges associated with large-scale wind turbines

Bigger, lighter and more flexible wind turbines rotors make the dynamic of the structure of a great importance. Scaling up the size of the machine is not an easy task. With the increase of wind turbine size, aeroelastic problems have been experienced on some wind turbines. Aeroelastic problems can end with structure collapse, therefore it is essential that the design of the wind turbine avoids aeroelastic instability. In general, the associated problems with the increasing of turbine size can be summarized as follows:

**Higher blade flexibility:** The continuing increase of wind turbine blade length makes the latter more flexible. Lighter flexible blades result in higher deformation, blade fluttering and alter turbine performance. Blade fluttering increases pitch moment at the blade root and pitching system and causes instability problems which reduce the operational life of the wind turbine (Hansen et al., 2006; Ahlstrom, 2006).

**Transportation problem:** One of the critical problems that faces the multi-megawatt wind turbines is the transportation problem. As the tower gets longer, tower base diameter increases. Nowadays the dimensions of the wind turbines towers are almost reached the limits of Europe roads capacity (maximum 4 m height (Council Directive 96/53/EC, 1996)).

**Rotor-tower strike risk:** Longer blades need bigger rotor-tower clearance to avoid blade-tower strike. The IEC 61400-1 states that blade-tower should be at least 1.5 times the blade deflection, (IEC 61400-1, 2005). For large wind turbines, rotor-tower clearance is also achieved by shifting the nacelle forward to keep the minimum required safety clearance. However, shifting the nacelle will create additional moment at the tower foundation that must be considered in the tower design.



**Installation collapse risk:** As the turbines support structure becomes taller, the risk of its collapse during the installation process becomes higher. Leaving the long tower standing for a long time without completing the assembly of the wind turbine (e.g. due to a delay of the other components or bad weather conditions) increases the risk of tower collapse. The problem arises when the tower exposes to certain wind conditions in which the shedding vortices frequency (known as Von Karman vortices)

matches with the natural frequency of the tower. In this case the tower starts to vibrate violently leading to fatigue damages.

**Blade-tower interaction:** Despite that the effect of blade-tower interaction on an upwind wind turbine is less than a downwind one (Zhao et al., 2014), it is a very complex problem to analysis analytically due to the high nonlinearity behavior of the aerodynamic forces in the system. Chattot (2006) and Shkara et al. (2018) showed in their study that even for upwind wind turbines, the tower has a significant effect on the unsteady working conditions of the blades as a result of tower blockage.

The aerodynamic forces on the rotor and the support structure change frequently during blades rotation. Therefore, it is necessary to design the turbine structure in such a way that the natural frequency of the system does not interface with the operating load frequency, so that tower resonance can be avoided. According to Danish standard DS472 (2009), simple statics analysis can be used for limited rotor size (up to 25 m or 200–250kW rated power). For larger wind turbines, accurate aeroelastic models involving detailed flow simulation and structure response are essential (DANSK STANDARD DS 472;

Rauh and Peinke, 2004; Tavner et al, 2007).

### 1.3 Related literature

Blade-tower interaction has been studied by many researchers with different methods in terms of level of the details and computational cost. The nonlinear vortex correction method with time-marching free wake has been adopted by Kim et al. (2011) to investigate the interaction between the tower and the blade. Their model showed a change in the normal force

coefficient by approximately 10 % of the average. They found that the influence of the tower radius variations on the interaction is bigger than tower clearance variations. Tang et al. (2017) developed an aeroelastic method to study the response of a 1.5-MW wind turbine by coupling a multibody method with a free vortex wake (FVW) method. The simulation results indicated that the aeroelasticity of a blade has significant effects on the wake geometries and structural responses. Flexibility of the tower can cause higher power and loads fluctuations than the blade, which can considerably affect the blade fatigue life design.

Furthermore, Lackner et al. (2013) investigated blade-tower interaction using potential flow that includes 2D and 3D versions. The drawback of their model was the inability to predict the flow field accurately as the flow over the tower encounters some viscous separation causing more complex flow.

On the other hand, Janajreh et al. (2010) performed a 2D CFD simulation of a downwind wind turbine to investigate the blade-tower interaction during the intrinsic passage of the rotor in the wake of the tower. The time history of the pressure, lift

and drag coefficients and the moments were evaluated for three different cross-sectional towers and compared with Panel method. The simulation results showed a reduction between 5 %, and 57 % of the aerodynamic lift forces during blade passage in the wake of the symmetrical airfoil tower. Following the same concept, the 2D simulation of Gomez and Seume (2009) of an upwind wind turbine showed a change of the stagnation point and the vortex separation points on the tower three times per



revolution. The 3D CFD simulation of Wang et al. (2012) showed a small influence of the tower on the aerodynamic performance of an upwind wind turbine. Results indicated that rotation of the blades in front of the tower will induce an obvious cyclic pressure drop and a noticeable flow separation from the tower due to the strong blade tip vortices.

Hsu and Bazilevs (2012) performed a 3D FSI simulation of full-scale upwind wind turbines. In their model the interaction

between the flexible rotor and the rigid tower of the three-blade 5 MW wind turbine showed a blade aerodynamic torque drop of 10 % – 12 % when it passes by the tower. In addition, a blade tip fluctuation of about 1 m is noticed. Moreover, the full CFD/CSD model of Carrion et al. (2014) showed that due to the proximity of the rotor to the tower, a deficit on the thrust and torque were observed on the NREL Phase VI wind turbine. In addition, the maximum deflections of the blades were observed after the blades passed the tower with 20 to 40 degrees at wind speeds of 7 and 20 m/s respectively. At 20m/s, the torque on

the elastic blades showed a 13 % increment from the rigid ones, which was attributed to the rapid blade oscillation. Furthermore, Yu and Kwon (2014) performed a loosely coupled CFD-CSD simulation of the NREL 5MW reference wind turbine. Results showed that due to the blade deformation, the blade aerodynamic loads are significantly reduced. In addition, the aerodynamic loads are abruptly dropped as the blades pass by the tower resulting in oscillatory blade deformation and vibratory loads, particularly in the flapwise direction.

## 2 Numerical model

The developed wind turbine simulation tool consists of three solvers: the CFD solver to predict the aerodynamic load, the FE solver to compute structure response and the dynamic mesh solver to update the grid position. The coupling between the fluid solver and the structure solver is implemented based on the partitioned approach, where each solver works independent from the other.

Modelling a complete aeroelastic wind turbine poses a huge number of challenges, namely the relative motion between the rotor and the tower. The choice of using the commercial software Ansys has been made taking into account the advantage of stability and the availability of Multiphysics tools in the software. In the following sections, the wind turbine specifications, flow and structure solvers and the coupling approach will be presented.

### 2.1 Wind turbine specification

The simulation is performed for a 5 MW upwind horizontal axis wind turbine. The specifications of the wind turbine are given in the Table 1. The wind turbine is equipped with three NREL 5MW blades, each blade has varying DUxx and NACA64 airfoils series along the blades span. The blade has a maximum chord length and twist angle of 4.65 m, 13.3° respectively (Jonkman, 2009). In order to simulate the flexible turbine and to simplify grid generation process, some modification of the original turbine design has to take place. The hub geometry is approximated to simple cylindrical shape and its diameter is

slightly increased, the blades roots have been cut so that the blades are not anymore physically attached to the hub. Finally, nacelle geometry is not considered in the simulation model (its weight is considered in the model). The reason behind these



changes will be discussed in the next section, nevertheless, the aerodynamic or structure effects of these changes are expected to be rather small.

**Table 1.** Wind turbine specifications.

| Blade | | |
|---|---|---|
| Length (w.r.t. Root Along Preconed Axis) | 61.5 | m |
| Mass | 17,740.0 | kg |
| First Mass Moment of Inertia (w.r.t. Root) | 363,231.0 | kg m |
| Second Mass Moment of Inertia (w.r.t. Root) | 11,776,047.0 | kg m2 |
| Rated tip speed | 80.0 m | ms-1 |
| Rotor | | |
| Orientation | Upwind | - |
| Configuration | 3 | blade |
| Diameter | 126 | m |
| Mass | 100,000 | kg |
| Shaft tilt | 6 | deg |
| Precone | 2.5 | deg |
| Hub | | |
| Diameter | 3 | m |
| Mass | 47,000 | kg |
| Height above ground | 115 | m |
| Nacelle | | |
| Mass | 130,000 | kg |
| Tower | | |
| Flange mass | 29,600 | kg |
| Tower mass | 361,300 | kg |
| Height above ground | 112 | m |
| Head diameter, thickness | 3, 0.02 | m |
| Base diameter, thickness | 5.5, 0.044 | m |
| Operation | | |
| Rated power | 5 | MW |
| Rated tip speed ratio | 7.55 | - |
| Cut-in (@ 6.9 rpm) | 3 | ms-1 |
| Rated (@ 12.1 rpm) | 11.4 | ms-1 |
| Cut-out | 25 | ms-1 |

## 2.2 Flow solver

The Navier-Stokes (NS) equations are solved in three dimensions for an incompressible flow using the commercial software Fluent, (ANSYS, 2018). Fluent is a general fluid dynamics software integrated into ANSYS Workbench which is an engineering simulation tool provided by ANSYS. The NS equations are discretized in the domain by means of finite volume method, where the applied mathematical conservation equations (mass, momentum and energy) are solved separately. The SIMPLE algorithm solves the pressure and the momentum equations in a predictor-corrector fashion. The convective flux is computed using the Second order Upwind Differencing Scheme (SUDS) in which the viscous term is discretized with the second order central difference scheme (ANSYS, 2018). As the flow is strongly turbulent near the rotor, the k-ω SST turbulent



model is adopted. It is considered as one of the most accurate turbulent models in the RANS class to predict the turbulent viscosity.

## 2.3 Structure solver

The dynamic response of the flexible wind turbine model is computed in the Transient Structural solver of Ansys (ANSYS 2018). The software uses the finite element method to solve the set of the partial differential equations of the equation of motion which can be written after assembling the finite elements matrices and vectors as:

$$M\ddot{x} + C\dot{x} + Kx = F_g + F_c + F_{Aero} \tag{1}$$

Where M, C and K are the mass, damping, and stiffness matrices respectively, $F_g$, $F_c$, and $F_{Aero}$ refer to the external load acting on the wind turbine structure due to the gravitational, centrifugal and aerodynamic forces respectively and x is the nodal displacement vector (Öchsner and Merkel, 3013). The aerodynamic force ($F_{Aero}$) is provided from an external module, where in this case, the aerodynamic forces are calculated in the CFD solver.

## 2.4 Dynamic grid solver

To take into account the motion of the structure in the CFD domain, the computational grid has to move according to the motion of the structure in both the space and time domains. An appropriate dynamic mesh method is necessary to avoid re-mashing high computational cost process and to ensure an efficient, robust and smooth grid motion. The adopted dynamic mesh solver in this model is based on the diffusion method, where the motion of the grid is governed by a diffusion equation:

$$\nabla(\gamma\nabla\vec{u}) = 0 \tag{2}$$

Where, $\vec{u}$ is the mesh displacement velocity and $\gamma$ is the diffusion coefficient (ANSYS, 2018). The boundary condition of the deforming surfaces is defined such that the mesh motion is tangent to the boundary (that is, the normal velocity component vanishes). The Laplace equation describes the motion of the boundary in the CFD computational grid and controlled by the diffusion coefficient. A constant diffusion coefficient refers to a uniform diffusion of the boundary motion through the grid.

In this model, the diffusion coefficient is set as a function of the boundary distance such that the high diffusion regions in the vicinity of the moving boundaries tend to move together. As a result, the refined cells height, growth ratios and quality near the structure surfaces are preserved.

## 2.5 Coupling approach

As the clearance between

As the clearance between the blade and the tower is of great interest, it is important that the flow solver sees the new position of the deformed blade. Therefore, the strong couple method is adopted in the simulation model. The procedure of the CFD-CSD analysis is presented in Fig. 1.



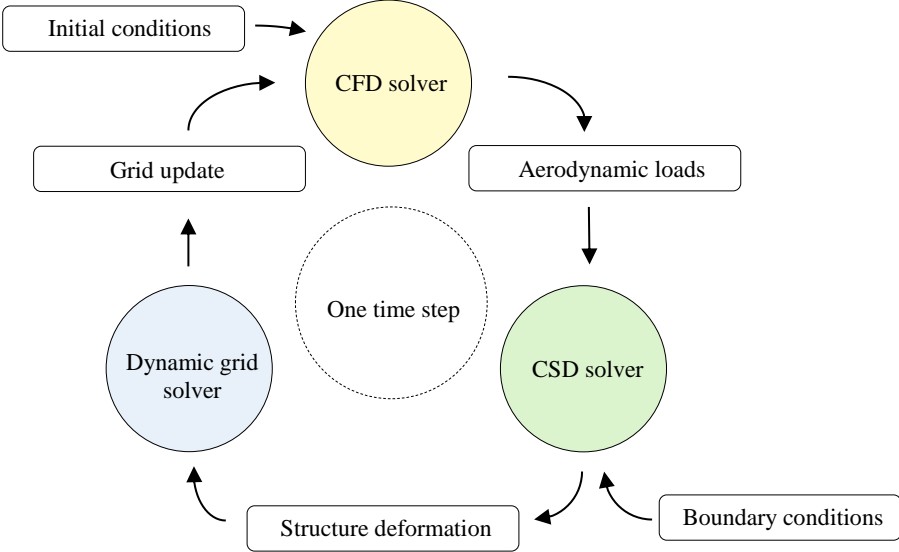

**Figure 1.** CFD-CSD coupling scheme.

The simulation starts with non-deformed structure, the flow solver computes the velocity and pressure distribution in the computational domain. Once the quasi-steady solution converges, the aerodynamic load is transferred to the CSD solver to compute the structure deformation. The new position of the deformed structure is then provided back to the CFD solver by updating the grid using the dynamic grid solver.

In the next time step, the aerodynamic load in the CSD solver is calculated taking into account the difference between the current load and previous coupled iterations as:

$$F_{CFD}^n = F_{CFD}^{n-1} + (F_{CSD}^n - F_{CSD}^{n-1}) \tag{3}$$

The coupling between the two computational domains is done by assigning each element in the flow domain to the nearest structure node in the structure domain in process known as mesh mapping. Hence, the predicted forces and moment on each cell face in the fluid domain is projected onto the finite element nodes in the structure domain.

**2.6 CFD Computational domain and grid generation process**

The computational domain has a rectangular shape where the turbine model positioned in the middle. The inlet and the outlet are placed 3D upstream and 3.5D downstream respectively and the sides are 2.5D each from the turbine geometry, Fig. 2. To simplify the grid generation process, the wind turbine geometry and its domain are segmented into five separated sections where each flexible component (i.e. the blades and the tower) have their own domains. This design is necessary to allow the deformation and motion of the wind turbine structure and to avoid grid elements collapse. A block structured grids with various types of grid topologies are adopted to generate high-quality grid for each individual domain separately.





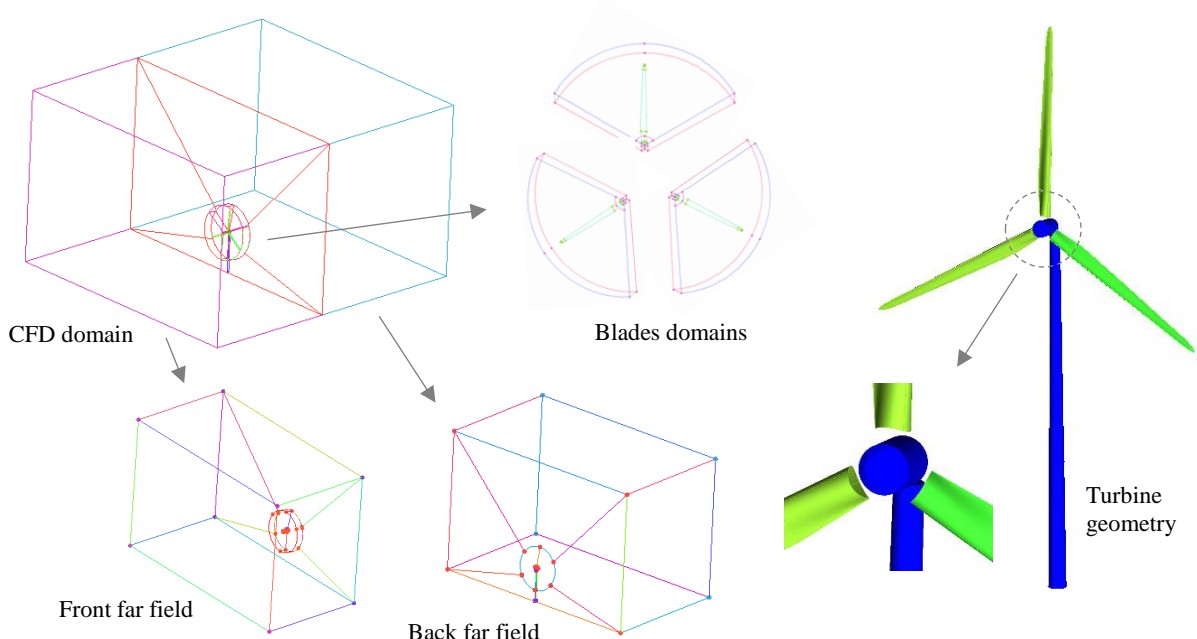

**Figure 2.** CFD domain and wind turbine geometry.

Wind turbine blades considered to be a complex geometry due to the thin, curved shape and big dimensions ratio. The mesh strategy for such a complicated system has a significant impact on the quality and accuracy of the results. As the structure deforms, the grid in the CFD domain has to be conformal to avoid elements high distortion. ANSYS ICEM CFD is one of the most advanced and powerful grid generation tools currently available. The software uses multi-block strategy to obtain high control of cells shapes, distribution, size and accurate fitting of the geometry. The structured grid in ICEM CFD consists of pure hexahedral elements. This kind of mesh is difficult to generate for complex geometries since the grid lines should not cross each other. On the other hand, it provides very good grid quality, which is essential for fluid-structure interaction (FSI) applications.

The blade domain is one third cylinder with an inclined surface to the back allowing the blade tip more space to deform in the flapwise direction, Fig. 3. Grid generation process starts with creating an initial block then segmenting it to smaller blocks, where their vertices, edges and surfaces are associated to the blade geometry to adopt the shape of the blade. The blocking strategy that used in the blade domain is consist of C-grid and H-grid. C-grid is used to capture the airfoil shape and create the refined high-quality boundary layers around the blade surfaces while the H-gird is set for the rest of the domain (Lecheler, 2009). To avoid elements collapse problems resulting from blade deformation, the blades roots have been detached from the





hub surface by cutting 1.5 m of the roots. Hence, each blade is placed in its own domain without having contact with the domain surfaces.

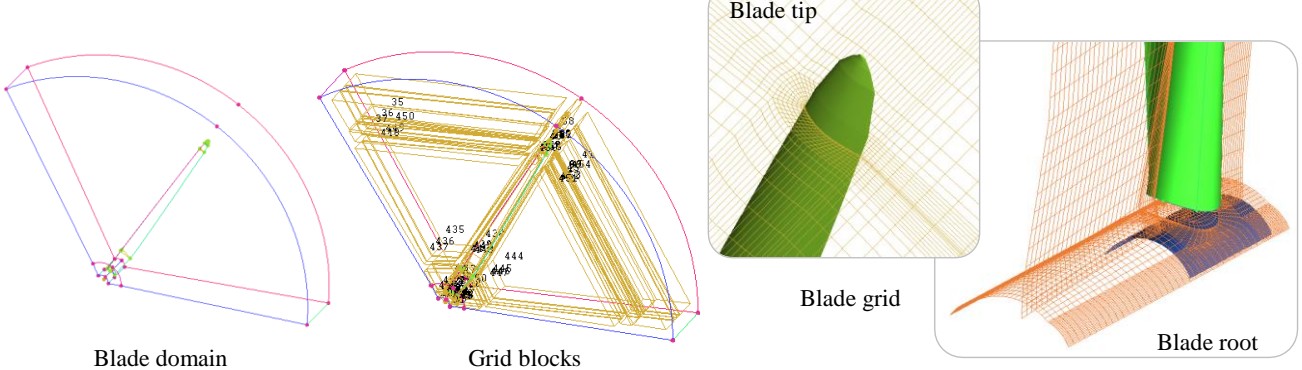

**Figure 3.** Blade computational grid.

The rectangular far filed domain is further segmented into two sections, front and back, Fig. 4. The front far field is the simplest part of the domain as it has no flexible bodies. The three blades domains are placed at the inner end of the front far field, therefore it has to feature a non-meshed space at the rotor position. The last domain is the back far field which includes the tower. The grid in this domain consists of an O-grid type surrounding the tower surface and H-grid for the rest of the

10 domain (Lecheler, 2009). The coupling between the rotor and the tower is done by using the non-overlapping sliding interface approach so that it's possible to rotate the blades keeping the tower stationary.

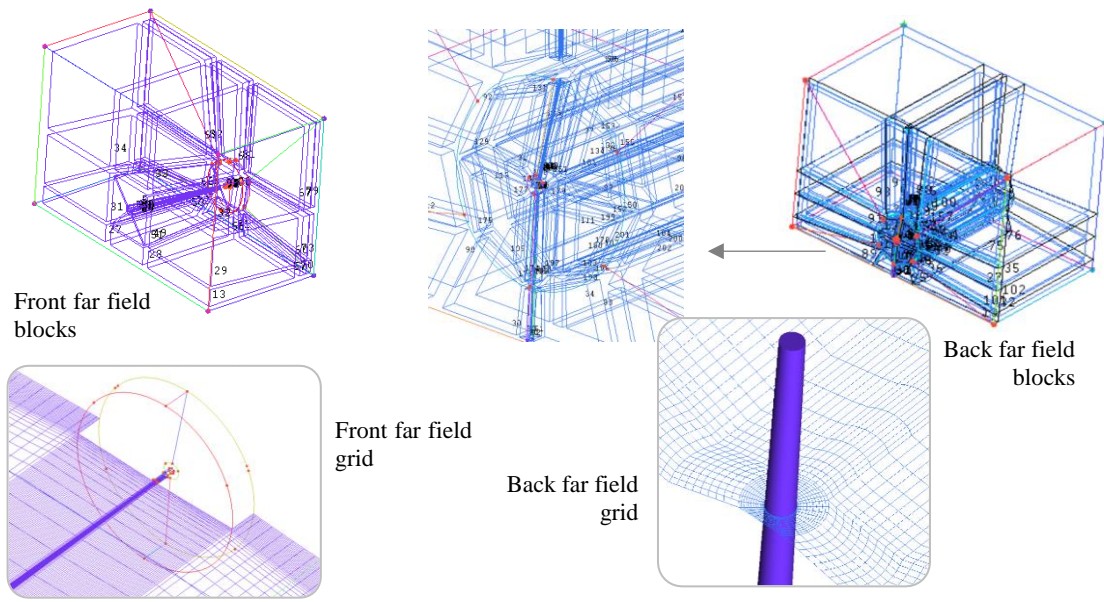

**Figure 4.** Front and back far field grid.





At the end of grid generation process, a total of 295 structure blocks are created to generate about 3 Million elements. Based on the computational domain grid generation strategy, the nacelle has been removed to avoid grid collapse due to the small distance between the nacelle and the back far field interface surfaces. The implemented structure blocking strategy comes

out with a suitable compromise among mesh size, grid resolution and cells quality.

## 2.7 Wind turbine structure model

Modelling of the wind turbine tower in the structure solver is very simple as the tower geometry is considered to be a simple cylinder structure. On the other hand, the presentation of the blades structure is quite challenging as the blades are made of numerous composite layers. To simplify blade structure presentation, the blades are modelled as a reduced equivalent beam

using the classical beam element theory (Thomson, 1966; Quaranta et al., 2005). The simple multibody approach models the blade as a series of rigid sections hinged and linked together with springs and dampers to represent structure stiffness and damping respectively. The beam model is computationally efficient as it reduces the number of DOFs and provides an accurate blade deformation. Each blade surface is segmented into 20 sections along the blade span and the flapwise, edgewise and torsional stiffnesses and damping coefficient are defined, Fig. 5.

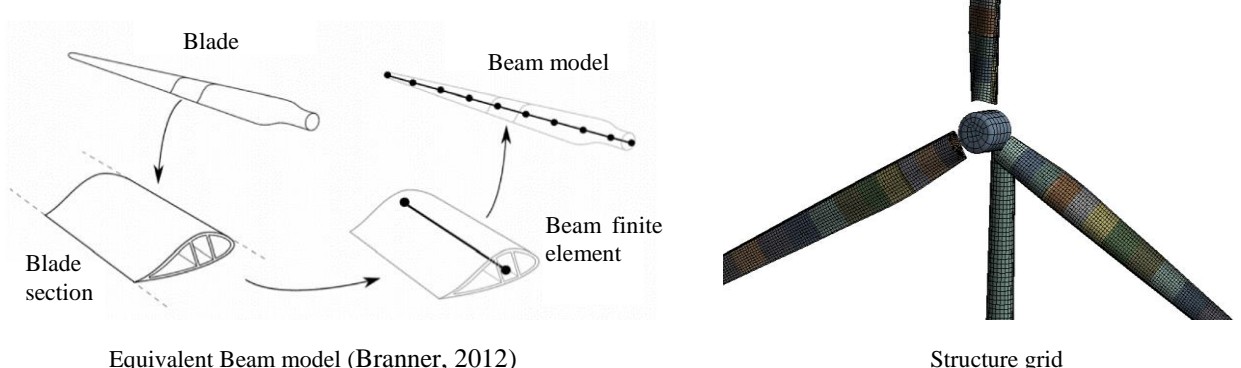

Equivalent Beam model (Branner, 2012)                    Structure grid

**Figure 5.** Wind turbine structure.

The wind turbine structure has been discretized with triangular and rectangular shell elements each has three and four nodes respectively, where each node has six-DOFs, three global translations and three global rotations. The final model has a total

of 27.5 thousand elements that represent a sufficient elements size to provide grid independency solution.

## 2.8 Simulation setup

As Ansys does not support rotation of the structure when it's coupled to the CFD solver, the simulation of the flexible wind turbine model is done in two steps. First the simulation is performed for flexible blades keeping the tower rigid. In this case,



the blades are rotating in the CFD domain while the structure solver computes the deformation of the individual stationary blades. Using this approach, the gravitational force is not possible to be considered in the structure model, therefore it has not been included. However, the centrifugal force due to the rotor rotation has been taken into account as it represents a radial force independent from the blade position. The deformations of the blades are computed based on the aerodynamic load of the

CFD solver. The forces and moments of the rotor are recorded from the CFD domain at the position of the tower head during the simulation time for the second simulation step.

In the second simulation step, the simulation of the same case is repeated, but this time the tower is considered to be flexible. The forces and moments that have been recorded from the first simulation step are set at the tower head. The rotor position in this simulation case is shifted with a mean tower deformation to the back so that the distance between the blades

and the tower is approximately conserved. Running the simulation for the second step allows the tower to see the flexible blades rotating in front of it in the CFD domain and to feel blades vibration as the loads are placed at the tower head from the first simulation. Using this approach, the rotor will not feel the vibration of the tower as they are not connected physically. The transient FSI simulation is performed for the following operation conditions in Table 2.

**Table 2.** Simulation boundary conditions.

| | | |
|---|---|---|
| Simulation type | Transient | - |
| Turbulence model | k-ω SST | - |
| Blades pitch angle | 0 | deg |
| Yaw angle | 0 | deg |
| Rotation speed | 12.5 | rpm |
| Inflow at hub height | 11.4 | ms-1 |
| Inflow turbulence intensity | 0 | - |
| Outflow | 0 | Pa |
| Ground | no slip wall | - |
| Turbine geometry | no slip wall | - |
| Interface surfaces | interface | - |
| Upper and sides boundaries | symmetry | - |
| Time step | 0.02 | sec |
| Total simulation time | 55 | sec |
| Dynamic grid diffusion coefficient | 1.5 | - |

The flexible tower in the second simulation case is fixed to the ground at the bottom and the hub is considered as a rigid rotating body in both simulation steps.

## 3 Results and discussions

After performing the first simulation step (flexible blades and rigid tower), the forces and moments of the rotor are averaged

for the last four cycles and set at the tower head. The static simulation of the tower showed a tower head mean deformation of about 0.88 m or 0.79 % of tower length and -0.003 m downstream and to the side respectively. Based on the new position of





the tower head the second simulation step (flexible blades and tower) were run after shifting the rotor to the new mean tower displacement position.

### 3.1 Aerodynamic performance

### 3.1.1 Tower forces

5    The motion of the blades in front of the tower will deflect the wind causing change of the stagnation point on the tower front surface. The tower suffers pressure drop three time per revolution known as 3P oscillations for three blades rotor. Fig. 6 shows the aerodynamic forces on the tower for one third rotor revolution, where 0º is when the blade in front of the tower. Each force component is plotted in percentage of its maximum value. The maximum normal force drop occurs after the blade pass's the tower with few degrees as the blade shadow reaches the tower. A maximum of 14.85 kN normal force is obtained on the tower

10   over one third rotor revolution. As the blade reached the tower a drop of about 52 % of the normal force is observed. The numerical model has been validated with a wind tunnel test of a scaled model. The pressure on the front surface of the tower has been recorded over the time by means of pressure sensors. Results showed a good agreement between measurements and the numerical model; more details about the test can be found in Shkara et al, (2017).

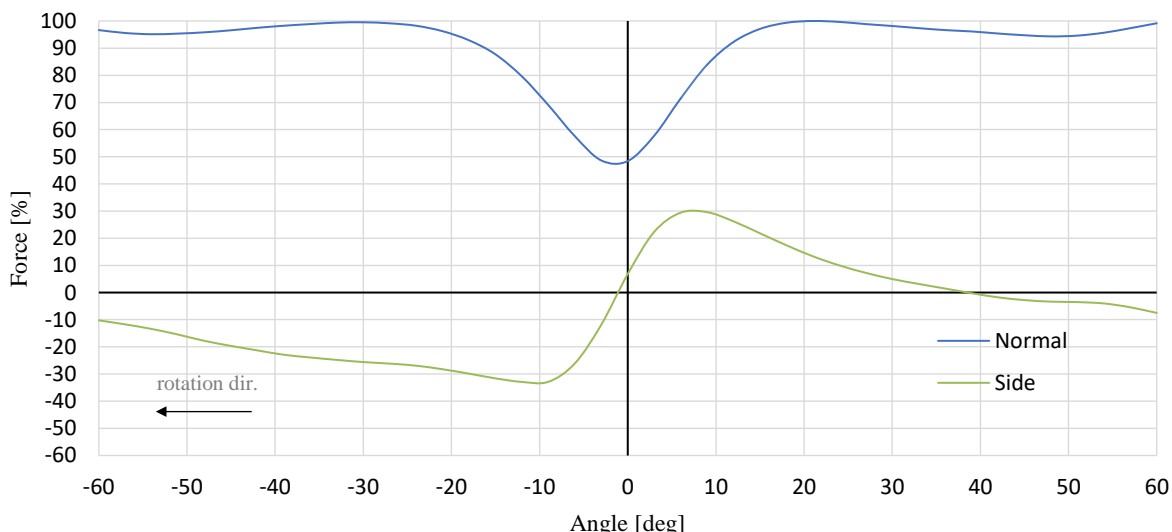

**Figure 6.** Aerodynamic forces on the tower for 1/3 rotation.

Furthermore, passage of the blades in front of the tower induces a side force fluctuation in short time. A maximum of 5.37 kN is observed on the tower which represent ± 30 % of the maximum normal force. These forces are caused by bound vortex circulation of the blades that disturb flow streamlines on both tower sides.



### 3.1.2 Rotor thrust

The effect of the blade-tower interaction is not only restricted to the tower, but the blade itself suffers aerodynamic impulsive forces as well. An individual blade thrust drop of about 3.1 % or 6.2 kN is noticed as the blade passes in the tower shadow. Fig. 7 shows the thrust distribution of the blade that pass's in front of the tower and the thrust of the complete rotor for one third rotation. In general, for the mentioned simulation conditions, a total rotor thrust drop of about 2.3 % three times per revolution is observed.

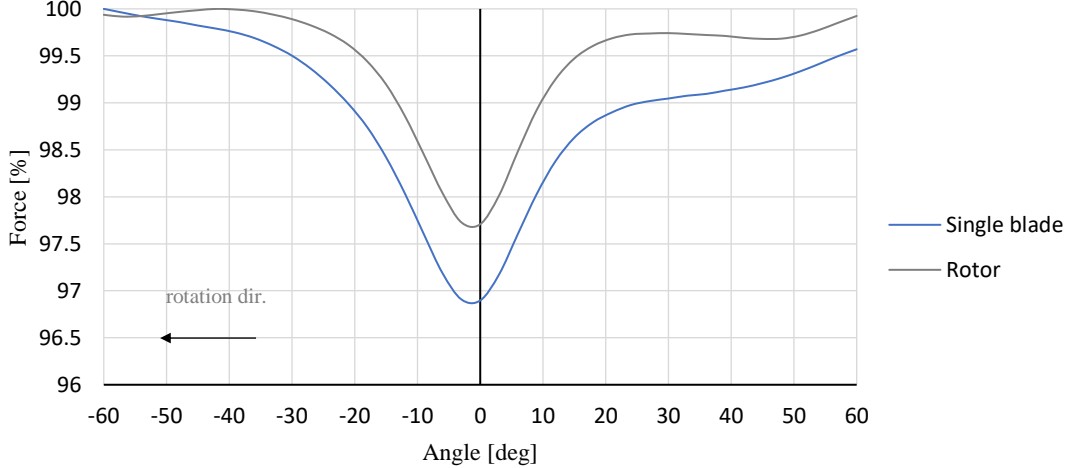

**Figure 7.** Single blade and rotor thrust for 1/3 rotation.

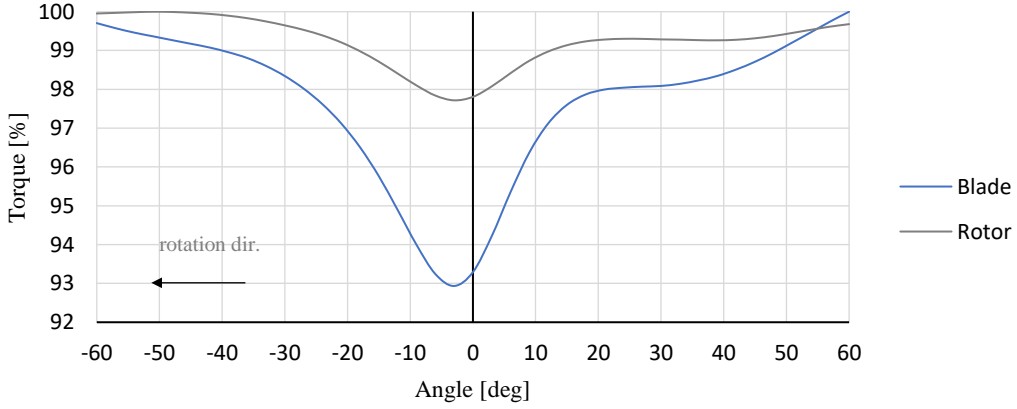

**Figure 8.** Single blade and rotor torque for 1/3 rotation.



### 3.1.3 Rotor torque

Fig.8 shows the generated torque of a blade passes in front of the tower and the total torque of the rotor for one third rotor revolution. The influence of blade passage in the vicinity of the tower results in a sudden decrease of the blade lift force which consequently causes a raped decrease of blade torque. An individual blade toque drops of about 67.8 kN.m or 7 % is observed as the flexible blade passes in front of the tower. Furthermore, a rotor toque drop of 66.5 kN.m or 2.3 % is occur three times per revolution.

The results of this simulation are in good agreement with Früh et al. (2008) simulation, moreover, similar flexible blades torque behavior is reported by Gebhardt and Roccia (2014). The 2D analysis of Früh et al. (2008) showed that movement of the blade in front of the tower will not only create effective velocity pulse, but it results in sharp change of the angle of attack of around 10%. Becker, (2017) showed in his CFD-CSD model of the NREL 5 MW that due to the blade elasticity, the torque deviation increased with respect to the rigid blade assumption. The effect of torsional deformation has been investigated by Yu and Kwon (2014) for the same simulation conditions (except wind profile). In their model, 6% rotor torque drop is noticed when the blades are considered to be flexible.

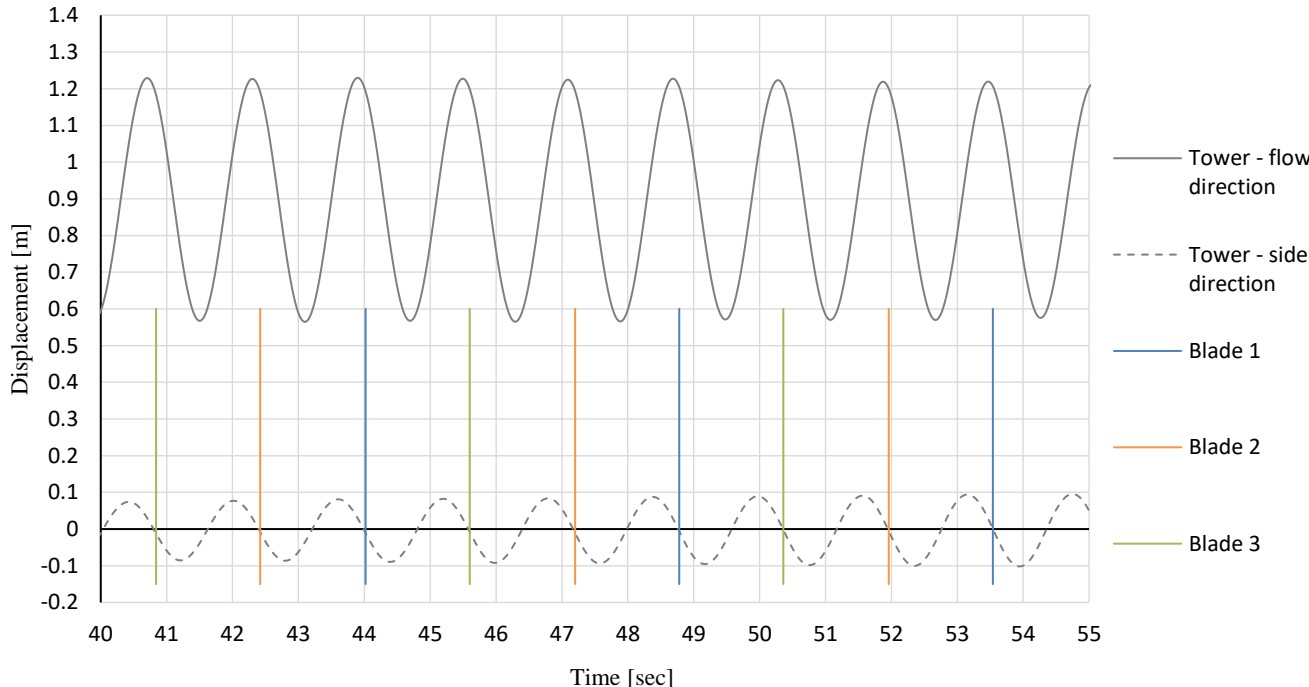

**Figure 9.** Tower displacement.





## 3.2 Structure dynamics

### 3.2.1 Dynamic response of the tower

Fig. 9 shows the displacement of the tower head in both downstream and side directions for the second simulation step. The vertical lines refer to the time point when the blades are positioned in front of the tower. The interaction between the rotor and

5   the tower can be seen clearly in the displacement of the tower in both directions. A tower oscillation of $\pm 32.5$ cm or $\pm 36.9$ % of the mean deformation and $\pm 0.095$ cm or $\pm 10.8$ % of the mean deformation downstream and to the sides are observed respectively. The tower is vibrating with a frequency of about 0.625 Hz which represents one third rotation of the rotor in the time domain.

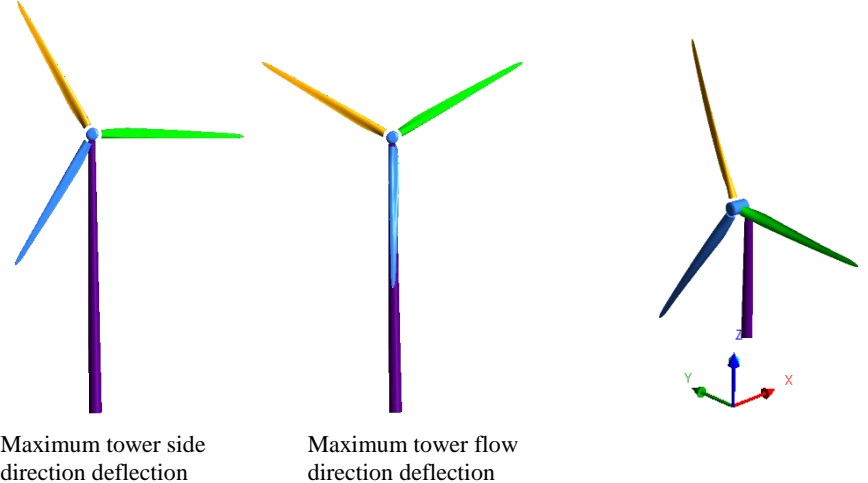

Maximum tower side          Maximum tower flow
direction deflection          direction deflection

**Figure 10.** Rotor position for the maximum tower deflections.

Although the complete wind turbine geometry is expected to experience less thrust due to the reduction of the tower projected area, still the maximum tower deformation in the flow direction occurred when one of the blades is located in front of the tower. The reason behind that is related to the azimuthal position of the other two blades, Fig. 10. At this time, the two other blades are located at the upper half sector of the rotor disc (above the tower head) resulting in higher bending moment

15   than the vertical blade and leading to further tower downstream displacement. That means, for these operation conditions (probably for different operation conditions as well), the azimuthal position of the rotor blades will have the dominant influence on the tower deformation than the blade that passes in front of the tower. Moreover, considering wind shear, rotation of the blade in the upper half sector will lead to increase their thrust force causing higher bending moment than the lower half sector.

Similar to the tower deformation in the flow direction, tower head side displacement is synchronized with the azimuthal

20   rotor angle as well. The side deformation of the tower in this case is resulting from the combination of the asymmetric rotor moment around the tower axis and the side component of the induced aerodynamic force caused by the blade rotation in front





of the tower. The maximum deflection of the tower in the side direction is observed when two of the blades are positioned on one side and the third blade is on the opposite side, Fig. 10.

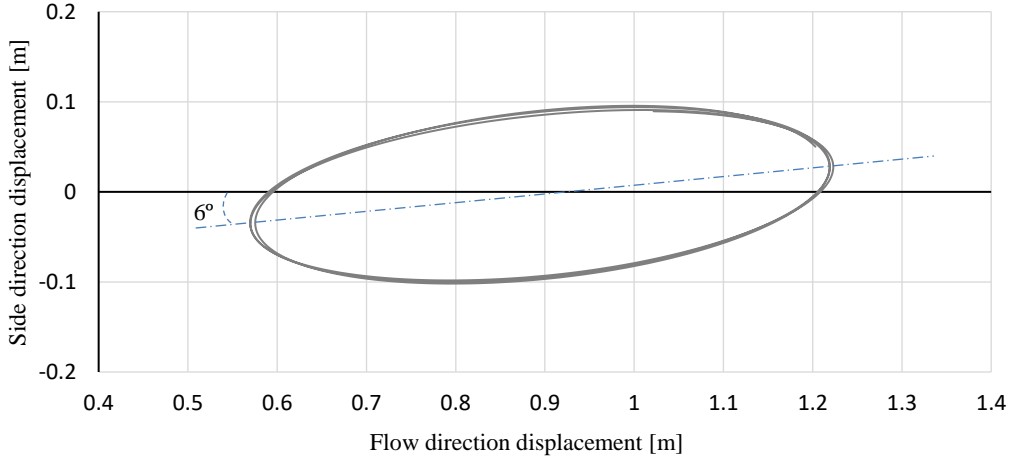

**Figure 11.** Tower head motion on a 2D plane.

The motion of the tower head for the second simulation step is plotted on a 2D plane in Fig. 11. The interface between the two displacements (flow and side directions) causes tower head motion following an elliptical pattern. The elliptic motion is inclined with an angle of 6° from the flow stream which is determined by the side displacement amplitude. This angle will be changed if the wind speed changed or the blades pitch angle changed.

### 3.2.2 Dynamic response of the blades

The flapwise displacements of the three blades over the time for the last 15 seconds are plotted in Fig. 12. A mean flapwise deflection of about 2.65 m is reached by the three blades which corresponds to 4.3 % of the blade length. The three blades oscillate with a phase shift of 120° from each other showing a coherence blades oscillation corresponds to the geometric layout of the blades in the rotor. The peak to peak deflection amplitude is about 16 cm which corresponds to 6 % of the mean deflection. Two main signals interfacing can be observed, the big amplitude with a frequency of 0.208 Hz is resulting from the

wind shear that tends to excite the blade vibration more than blade-tower interaction. Similar blade response has been noticed in the work of Yu and Kwon (2014) as the blades passed by a rigid tower and the model of Tang et al. (2017) when the blade passes a flexible tower in their combined vortex wake and multibody dynamics model. The biggest flapwise deflection occurs at about 225° from the tower position which is something expected as the blade is subjected to the highest thrust when the blade is at the top (highest wind speed). The minimum blade displacement is observed at about 15° after the blades passes

through the tower shadow.

The influence of the blade-tower interaction appears as a small dip in the displacement of the blades tip with an amplitude of ± 5 cm or ± 1.9 % of the blade mean deflection after the blade passes the tower. A response delay of about 0.5 seconds or



38° azimuthal angle is noticed. Similar lag time structure responses phenomenon has been observed by Tanget al. (2017) as well due to the aeroelastic effects.

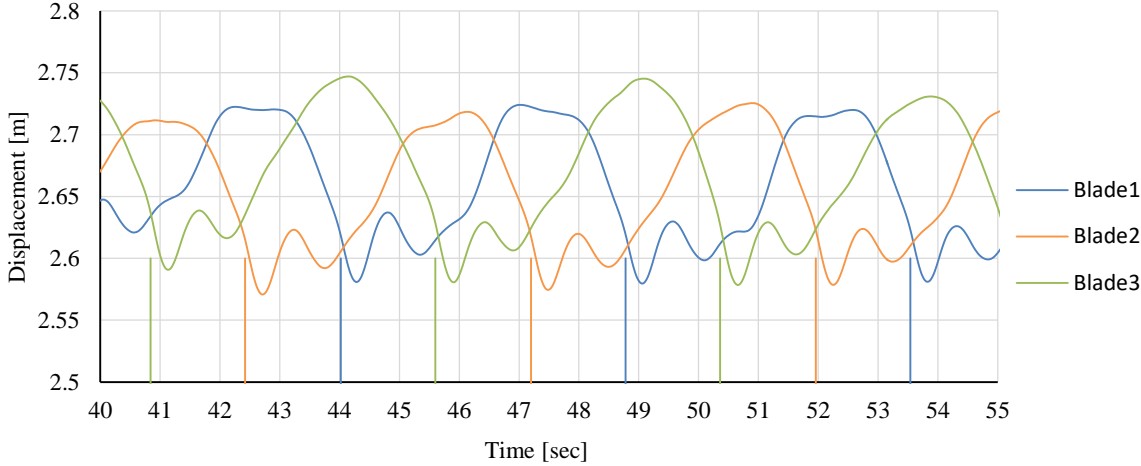

**Figure 12.** Blades flapwise displacements.

A CFD simulation of the same rotor and boundary conditions but with bigger tower diameter and pre-deformed blades based on BEM calculation has been performed by Shkara et al. (2018). In comparison to the deformation of the BEM method, the mean flapwise deflection of the current flexible model showed a higher blade flapwise deflection by 38 cm or 14 %. This indicates that CFD thrust force is slightly higher than the BEM method or vice versa. However, the difference is very small and has a rather neglectable influence on the wind turbine performance.

The obtained flapwise deflection is relatively small compared to what has been achieved in the previous publications (Jeong et al., 2013; Becker, 2017; Dose et al. 2018) for the same simulation conditions. The reason behind that is due to the consideration of the centrifugal force in the current model. Rotation of the blades creates a centrifugal force that can reach up to 8 G in magnitude causing increase of blade stiffness in both flapwise and edgewise directions and alter their natural frequency (Bertagnolio et al., 2002). As a result, blade deformation is considerably decreased compared to a stationary blade
subjected to the same load.

       The simulation of the same wind turbine model has been performed using blade element momentum (BEM) method and multibody dynamics approach for a rigid tower. Fig. 13 shows blades flapwise displacements of the CFD and the BEM models. It's clear that the BEM method predicted higher mean blade deformation than CFD. A mean blades flapwise displacements of 3.9 m is obtained using BEM method compared to 2.65 m using CFD which corresponds to a difference of 32 %. The response
of the blades as they pass in front of the tower shows very similar behavior for both methods (i.e. CFD and BEM). However,





the oscillation amplitudes of the BEM blades are bigger than the CFD. The peak to peak deflection amplitude is about 32 cm in the BEM model compared to only 16 cm in CFD which corresponds to 50 % lower blade deflection oscillation amplitude.

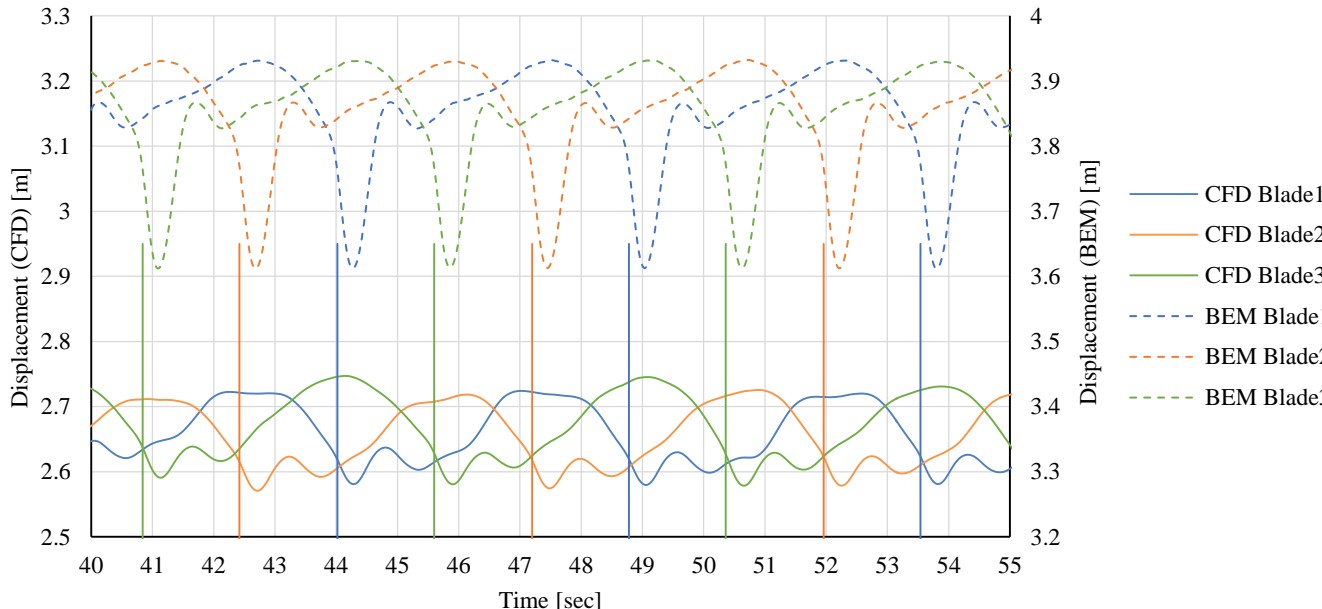

**Figure 13.** Blades flapwise displacements of the CFD and BEM models.

The reason behind the differences in the displacements is related to the fact that the aerodynamic damping is not considered in the BEM model which is part of the solution of the CFD. In addition, BEM method predicted higher rotor thrust than the CFD by about 22.4 %.

A mean deflection of about 10 cm is observed in the edgewise direction with an oscillation amplitude of about ± 1.5 cm or ± 0.6 % of the mean blade deformation, Fig. 14. The amplitudes of the edgewise displacements are very small which is due to

the fact that the gravity is not considered for the blades structure. Similar to the flapwise oscillation, the blades vibrate in the edgewise direction because of aerodynamic forces change over the azimuth angle (wind shear) and the interaction with the tower shadow. Furthermore, the blades vibrate because the turbulent nature of the flow over the blade profile although the income flow is uniform. Früh et al. (2008) showed in their study that the flow over a wind turbine is either fully turbulent as a consequence of the turbulent intensity in the atmospheric flow or the transition occur mostly at a distance of 10 % of the blade

leading edge.

Similar to the blades flapwiswe deflections, the blades edgewise deflections of the BEM model are higher than the CFD model. The BEM model showed a mean blade edgewise deflection of 40 cm with an oscillation amplitude of about ± 2.5 cm due to the passage of the blades in the tower shadow. The blades edgewise deflection is related to the blades toque which in the case of BEM model is higher than the CFD model by 19 %.

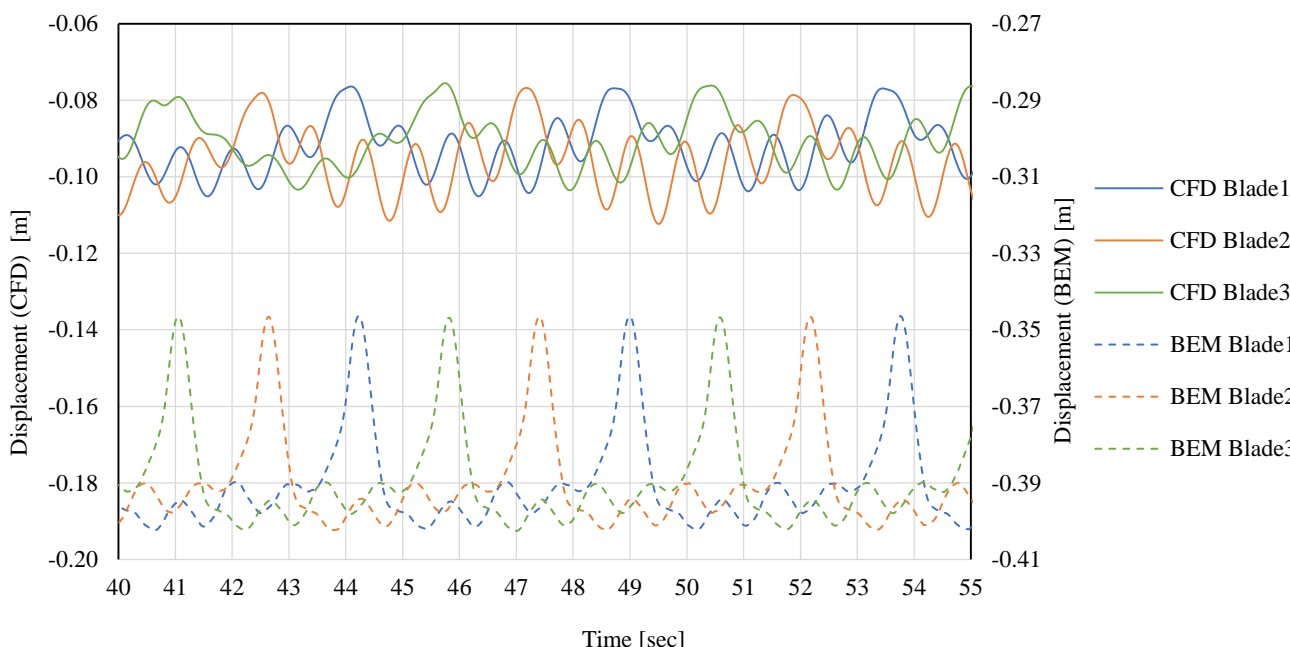

**Figure 14.** Blades edgewise displacements of the CFD and BEM models.

## 4 Conclusion

In this paper, a coupled CFD-CSD numerical simulation method is presented to investigate the dynamic response of a 5 MW

upwind wind turbine structure taking into account blade-tower interaction. The coupling between the fluid solver and the structure solver was implemented based on the partitioned approach. Both the blades and the tower are considered to be flexible for the nominal operation condition simulation. The results showed tower mean displacement of about 0.79 % of tower length downstream with an oscillation amplitude of ± 36.9 % and ± 10.8 % of the mean deflection downstream and to the sides respectively. The interaction with the tower causes blades oscillation in both flapwise and edgewise directions with a phase

shift of 120 degrees from each other. The highest deformations of the blades were dominant by the wind shear and the rotor azimuthal angle described the motion of the tower head. The influence of the blade-tower interaction appears as a small dip in the displacement of the blades tip with an amplitude of 1.9 % of the blade mean deflection and a sudden rotor torque drop of 2.3 % three times per rotation. The simulation of the same wind turbine model has been performed using blade element momentum (BEM) method with multibody dynamics approach for a rigid tower. The simulation results showed that BEM

model overestimate both rotor thrust and torque which resulted in higher blades flapwise and edgewise deflections and their





oscillation amplitudes. The additional cyclic aerodynamic loads on both the tower and the blades due to the blade-tower interaction induces fatigue loads which are considered to be essential for the structure lifetime prediction and analysis.

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
