# Peer review of "Aeroelastic Response of a Multi-Megawatt Upwind HAWT Based on Fluid-Structure Interaction Simulation"

_Wind Energy Science, 2019_

## Referee Comment (RC1) · Abdul Baseer (Referee) · 11 Jul 2019

The article deals with important problem that would be significant in the design of the future bigger wind turbines structures. The scientific quality of the manuscript is quite good. The abstract and the conclusion covers the content of the manuscript clearly and the sections are well structured. The manuscript covered sufficient literature. The results are clearly described and the discussion is well presented and emphasized with comparison with other published papers. However there are some additional clarity or corrections are required: What are the natural frequencies of the system? The biggest flapwise deflection occurs at about 225° from the tower position which is something

expected as the blade is subjected to the highest thrust when the blade is at the top (highest wind speed). What is the reason for the delay? Why is the amplitude deflection in BEM bigger than CFD?

––––––––––––––––––––––––––––––

---

## Referee Comment (RC2) · Anonymous Referee #2 · 11 Jul 2019

-page 6 line 24 remove:"As the clearance between" -figure 14 remove (CFD) from y-axis -capter :3.2.2 the reason why the tower shadow effect in the BEM is much higher compared to the CFD model should be described in detail.

---

## Short Comment (SC1) · 21 Jul 2019

The paper deals with a very important and interesting topic in wind energy. However, there are some issues that need to be solved before the final publication.

1. The level of English language is poor and makes portions of the paper incomprehensible.
   a. Please do not use more than one verb per sentence (for example is occur, is depend)
   b. It either is "to depend on" or "to be dependent on"; never is depend
   c. Blades domain / blades deflection → blade domain/ blade domain
   d. Wind turbines rotor → wind turbine rotor
   e. What is pass's? Do you mean passes?
   f. What is "Maximum tower side/flow direction deflection"?

   Please also add the missing "the" and missing prepositions.

2. cm is no good measurement unit. Please use m (10^-2m) instead.
3. Section 1: I missed the point on what is new in the present paper with respect to your previous publications, Could you please clarifiy?
4. I did not get the inflow conditions. Is it constant in height? Logarithmic law? Exponential law? Constant in time? Do you use veer? Please clarify.
5. Could you please provide the convergence histories of the blade and tower deflection over time? (Deflection over cycle number)
6. Would it be possible to add a table with the results of section 3.2.2 included?
7. In section 2 you mention that "modelling a complete aeroelastic wind turbine poses a huge number of challenges" but you name only one afterwards. Is it one or more than one?
8. Why did you choose the flow domain to be so small? They are usually ten times larger in each direction comprising more points.
9. Could you give more details on the grid? What is the resolution on the blade surface, of your boundary layer grid, boundary conditions on the blades and tower?

---

## Short Comment (SC2) · 23 Jul 2019

Thank you for your fast reply.
Please find below the reply to your answers,

- The answers to point 4 and 9 should be added to the paper since they are of major interest.

- Regarding point 6: No, I was thinking about a table containing the numbers which you discuss in the section 3 (maximum of normal forces, maximum of side forces of elastic and non-elastic tower; blade and rotor thrust or torque drop and so

on.) but especially section 3.2 (oscillation amplitude, mean deformation, vibration frequency, for tower and blades, respectively). It would enable a fast overview about the numbers and simplify the comprehension of your discussion.

- Regarding point 7: then either name them all or change to something like: the most prominent, among others, is the relative motion between the rotor and the tower.

- Regarding point 8: Thank you for the honest answer. Then it is ok for me.

- Please pay also attention to page six, line 24
* * *

---

## Author Response (AR1)

**Response to Referee Abdul Baseer:**

[Referee]    The article deals with important problem that would be significant in the design of the future bigger wind turbines structures. The scientific quality of the manuscript is quite good. The abstract and the conclusion covers the content of the manuscript clearly and the sections are well structured. The manuscript covered sufficient literature. The results are clearly descried and the discussion is well presented and emphasized with comparison with other published papers.

However there are some additional clarity or corrections are required:

What are the natural frequencies of the system?

[Authors]    For the current rotation speed, the first two natural frequencies are 0.231 and 0.233 Hz which are tower natural frequencies. The third to sixth natural frequencies are blades flapwise which are: 0.709, 0.826, 0.839 and 0.899 Hz. The seventh to the tenth are: 1.04, 1.059, 1.477 and 1.496 Hz which represent the blades edgewise natural frequencies. That means the operation conditions for this simulation is not in the resonance condition.

[Changes]    A table of the natural frequencies of the system is added (Table 3).

[Referee]    The biggest flapwise deflection occurs at about 225∘ from the tower position which is something expected as the blade is subjected to the highest thrust when the blade is at the top (highest wind speed). What is the reason for the delay?

[Authors]    The structure response of the blades and the tower are due to the structure inertia. For the current wind turbine model and simulation conditions, a blade flapwise delay response of 0.5 sec is noticed as the blade passes in front of the tower, similar delay is also noticed as the blade exposed to the higher wind speed as you mentioned at the top of the rotation circle.

[Changes]    The answer for this question is added to the discussion of the dynamic response of the blades.

[Referee]    Why is the amplitude deflection in BEM bigger than CFD?

[Authors]    Apparently BEM predicted higher thrust than CFD by 22.4%. That can be related to the fact that the BEM does not predict the thrust accurately in the case of flow separation or overestimate it which in this case occurs near the blades root. Furthermore, BEM can give only one constant value for a certain operation condition (as the method is based on the wind tunnel measured lift and drag coefficients), in contrast, CFD uses advanced turbulent models to predict the transient lift and drag forces of the blades which might be different from the previous rotation of the same blade position.

[Changes]    The answer for this question is added to the discussion of the dynamic response of the blades.

**Response to Referee #2:**

[Referee]    page 6 line 24 remove:"As the clearance between"

[Authors]    Thanks! Will be corrected in the final version.

[Changes]    Has been corrected in the manuscript.

[Referee]    figure 14 remove (CFD) from yaxis

[Authors]    To reduce the gap between the CFD and the BEM blades flapwise deformations curves, the deflection of the BEM blades have been shifted to the bottom, therefore we are showing results of the two different methods on two y-axes. They are needed to indicate which axis represent the results for which method.

[Changes]    No changes took place.

[Referee]    capter :3.2.2 the reason why the tower shadow effect in the BEM is much higher compared to the CFD model should be described in detail.

[Authors]    For the current simulation conditions, BEM predicted higher rotor thrust and torque than the CFD model. The blade structure behaves like a spring, the more you compress it the more the displacement amplitude will be. Therefore the displacement amplitude of the BEM blade is bigger than the CFD model as it passes in the tower shadow. (This paragraph will be added to the last version of the manuscript).

[Changes]    The answer for this question is added to the discussion of the dynamic response of the blades.

**Response to Annika Länger-Möller:**

[Reviewer]  The paper deals with a very important and interesting topic in wind energy. However, there are some issues that need to be solved before the final publication.

The level of English language is poor and makes portions of the paper incomprehensible.

[Authors]  Thank you for your interesting questions. English is not my mother language as you noticed, I will try to correct the mistakes.

[Changes]  The languge of the manuscript has been revised.

[Reviewer]  cm is no good measurement unit. Please use m (10^-2m) instead.

[Authors]  You are right, I'll change the unit in the final version.

[Changes]  The unit has been changed.

[Reviewer]  Section 1: I missed the point on what is new in the present paper with respect to your previous publications, Could you please clarifiy?

[Authors]  Additional section after "Related literature" can be added to clarify the objective of the manuscript. In short, the idea behind this work is to use the CFD as an aerodynamic tool to predict flow structure and to study the response of the wind turbine structure (namely the tower) due to blade-tower interaction. The previous publication focused on the effect of tower shadow on the blades and assumed rigid tower. In this work we introduced a flexible tower in addition to flexible blades. Using this method, the aerodynamic loads on the tower can be predicted with much more details than using the classical BEM method and consequently structure dynamics.

[Changes]  A new section (1.3 Objective) has been added to the manuscript.

[Reviewer]  I did not get the inflow conditions. Is it constant in height? Logarithmic law? Exponential law? Constant in time? Do you use veer? Please clarify.

[Authors]  Wind speed gradient (wind shear) has been considered at the inflow with a velocity profile following the power law function below:

[Figure]

$$V(z) = V_{\mathrm{m}} * \left(\frac{Z}{Z_{\mathrm{hub}}}\right)^{0.1}$$

Where, $V(z)$ is the velocity at any height, $V_m$ is the mean velocity (in this case 11.4 m/s), $Z$ is the height and $Z_{hub}$ is the hub height.

[Changes]    Inflow conditions have been declared in the "simulation setup" section.

[Reviewer]   Could you please provide the convergence histories of the blade and tower deflection over time? (Deflection over cycle number)

[Authors]    The deflection history of the blades over time are figures 13 and 14 for the flapwise and edgewise directions respectively. Tower deflection history is figure 9. Deflection over cycle number is quite clear from the position of the tower indicated by the vertical lines, where each three sequential vertical lines present one rotor rotation.

If you mean the history from the beginning of the simulation (i.e from time=0 sec.), blades flapwise deflections for example:

[Figure]

Tower deflections:

[Figure]

Where the vertical lines in the last figure refer to the time point when the blades are positioned in front of the tower. Please note that to avoid high structure deformation at the beginning of the simulation (due to the assumption of non-deformed structure at time=0 sec), the rotation velocity of the rotor is ramped up linearly from 0 to 12.1 rpm during the first 7 simulation seconds. As a result, the thrust force that leads to blade flapwise and tower streamwise deflections are distributed smoothly during the first 7 seconds, reducing the risk of grid collapse.

[Changes]   No changes took place as the results are already mensioned in the manuscript.

[Reviewer]   Would it be possible to add a table with the results of section 3.2.2 included?

[Authors]   if I understood your question correctly, you asked to add a tables for the blades deformations with time. The data in these tables would be huge as the sampling rate was 0.02 sec. it doesn't make sense to increase time step in the table as the curves won't be smooth anymore. Therefore we reduce the amount of these data by plotting them in the figures above.

[Changes]   No changes took place since all the inofrmation are given in the figeres and discussed in the text, adding tables will make it over-defined.

[Reviewer]   In section 2 you mention that "modelling a complete aeroelastic wind turbine poses a huge number of challenges" but you name only one afterwards. Is it one or more than one?

[Authors]   Yes there is more than one. We didn't go into details, but some other points can mentioned as well. The aerodynamic model should satisfy the following requirements, which are not easy to combine in one software:

•       Support more than one flexible body interacting with each other (rotor blades and the support structure).

- Provides an appropriate presentation of the blade structure as the blades have numerous composite layers making the calculation very computationally expensive.
- Should be able to operate in transient state so that the output can be used to compute the response of the structure in the time domain.

[Changes]    The points have been added to the manuscript.

[Reviewer]    Why did you choose the flow domain to be so small? They are usually ten times larger in each direction comprising more points.

[Authors]    For four main reasons:

1.    All of the simulations were run on a local computer with limited computational resources, therefore increasing the domain and cells sizes will increase computational time (the current model has been run for more than three weeks).
2.    We have limited software (Ansys) license, which mean we cannot use many processor cores (the current simulation was run using 3 cores).
3.    The aim of this work is to study the dynamic response of the structure and the near field flow structure, but not the far wake of the wind turbine for example. Therefore, there is no need for large domain downstream, also upstream as the incoming flow is laminar with 0 turbulent intensity.
4.    Increasing the domains size will increase the time to reach the quasi-steady state of the wind profile as the wind has to travel along the depth of the domain till the end of the domain (the simulation domain was initialized with the constant mean wind speed 11.4 m/s for all the cells).

        Nevertheless, we have made sure that the chosen domain size and the number of cells won't have influence on the results.

[Changes]    No changes took place.

[Reviewer]    Could you give more details on the grid? What is the resolution on the blade surface, of your boundary layer grid, boundary conditions on the blades and tower?

[Authors]    Each blade has 51 elements around the airfoil section and the tower has 40 elements around its section. The first layer is located 1 cm above blades and tower surfaces with a growth ration of 1.3. This is the minimum cell height we could achieve so that the dynamic grid solver can work without problems of grid collapse when the structure deforms. All wind turbine surfaces have been defined as no slip wall as mentioned in Table2.

[Changes]    Grid details have been added to the manuscript.

**Aeroelastic Response of a Multi-Megawatt pwind HAWT ased on luid-tructure nteraction imulation**

[revised manuscript text omitted]